# Single Center Experience with a 4-Week ^177^Lu-PSMA-617 Treatment Interval in Patients with Metastatic Castration-Resistant Prostate Cancer

**DOI:** 10.3390/cancers14246155

**Published:** 2022-12-14

**Authors:** Jukka Kemppainen, Aki Kangasmäki, Simona Malaspina, Bernd Pape, Jarno Jalomäki, Kalevi Kairemo, Juha Kononen, Timo Joensuu

**Affiliations:** 1Docrates Cancer Center, 00180 Helsinki, Finland; 2Department of Clinical Physiology and Nuclear Medicine, Turku University Hospital, 20520 Turku, Finland; 3Department of Mathematics and Statistics, University of Vaasa, 65101 Vaasa, Finland; 4MAP Medical Technologies Oy, 00180 Helsinki, Finland; 5Department of Nuclear Medicine, MD Anderson Cancer Center, University of Texas, Houston, TX 77030, USA

**Keywords:** prostate cancer, PSMA PET/CT, lutetium, radionuclide therapy

## Abstract

**Simple Summary:**

The optimal treatment regimen with ^177^Lu-PSMA-617 for metastatic castration-resistant prostate cancer patients is not known. In this retrospective analysis, the efficacy and impact of a four-week treatment interval on patient outcome and safety were investigated. A significant PSA response was observed in 58.7% of patients, and this was associated with better OS and PFS without compromising treatment safety. A shorter treatment interval may broaden the therapeutic window, especially in patients with rapidly progressing disease. Pre-treatment staging PSMA PET/CT was not helpful in identifying responders from non-responders. Therefore, better biomarkers are needed to aid in patient selection of potential treatment candidates.

**Abstract:**

Background: ^177^Lu-PSMA-617 is a promising theragnostic treatment for metastatic castration-resistant prostate cancer (mCRPC). However, both the optimal treatment dose and interval in mCRPC and the rate of identification of responders from non-responders among possible treatment candidates are unknown. Methods: 62 men with mCRPC who were treated with ^177^Lu-PSMA-617 during 1/2017–2/2019 were included in the study. Treatment responses, overall survival (OS) and progression free survival (PFS) were determined. The median follow-up time was 1.4 years (IQR 0.5–2.2). Tumor volume of metastases (MTV), SUVmax and tumor lesion activity (TLA) were quantitated from pre-treatment PSMA PET/CT images together with pre-treatment PSA. Results: An average of three treatment cycles (2–5) were given within a four-week interval. PFS was 4.9 months (2.4–9.6) and OS was 17.2 months (6–26.4). There were no major adverse events reported. A significant PSA response of >50% was found in 58.7% of patients, which was significantly associated with longer OS, *p* < 0.004. PSA response was not associated with staging PSMA-derived parameters. Conclusions: ^177^Lu-PSMA-617 treatment in four-week intervals was safe and effective. Almost 60% of patients had a significant PSA response, which was associated with better OS. Pre-treatment PSA kinetics or staging PSMA PET/CT-derived parameters were not helpful in identifying treatment responders from non-responders; better biomarkers are needed to aid in patient selection.

## 1. Introduction

During the past decade of theranostics research, prostate-specific membrane antigen (PSMA)-targeted internal radiotherapy with ^177^lutetium-labeled ^177^Lu-PSMA-617 or ^177^Lu-PSMA-I&T has been under keen scientific interest due to the limited number of therapeutic options for mCRPC patients [1,2]. Possible candidates for treatment should be evaluated with PET/CT imaging using ^68^Ga- or ^18^F-labeled PSMA targeting tracers. The intensity and distribution of PSMA uptake guides the assessment of treatment eligibility, although definitive uptake criteria for patient selection have not been established [3,4].

The safety and efficacy of Lu-PSMA treatment have been demonstrated in numerous retrospective and prospective studies [5,6,7,8,9,10,11,12,13,14,15,16,17]. Across studies, the proportion of individuals with a significant prostate specific antigen (PSA) response of >50% ranges from 32% to 64%. However, the optimal treatment regimen for patients is not known. Open questions include: “should dosing be individually adjusted or fixed?” and “what is the optimal treatment interval?”. Rathke et al. demonstrated that patients receiving the highest treatment dose achieved the highest rate of partial remission. Specifically, seven out of ten patients in the 9 GBq group vs. two out of ten in the lowest 4 GBq dose group achieved partial remission, although a significant correlation between treatment dose and PSA response could not be demonstrated [18]. Treatment intervals varied from six to twelve weeks among the first retrospective trials. More recently, studies have been performed with a shorter six-week treatment interval. The large prospective VISION trial with more than 800 patients was performed with a fixed 7.4 GBq dose and a six-week treatment interval. Seifert et al. compared the impact of a higher 7.5 GBq dose with a shorter six-week treatment interval (n = 41) to a lower 6 GBq dose with an eight-week interval (n = 37). There was a trend towards a higher number of responders and a longer survival time without compromising treatment safety in the more intense 7.5 GBq dose group with six-week treatment interval, although the difference was not statistically significant [13].

It is known that some mCRPC patients who do not respond to the first PSMA targeted radioligand therapy (PSMA-RLT) might respond after one to two subsequent treatments [19]. In addition, a PSA-flare-related increase in PSA after the first treatment might obscure the early response assessment. The number of non-responders was up to 29% in VISION trial [17], which suggests that a six-week treatment interval might not be intensive enough for some patients. Since PSMA-RLT has a good safety profile, shorter treatment intervals can be explored. It is not known whether three- or four-week treatment intervals would induce more responses and improve the outcome of PSMA-RLT, especially among mCRPC patients with high volume disease and rapidly rising PSA. 

Our aim was to retrospectively assess the impact of a four-week treatment interval on patient outcomes and to evaluate any possible adverse events related to treatment toxicity. Moreover, we wanted to investigate whether pre-treatment staging PSMA PET/CT could provide prognostic information or aid in identification of those subjects who are likely to benefit most from PSMA-RLT. 

## 2. Materials and Methods

### 2.1. Subjects and Study Design

This is a single-center retrospective analysis in 62 men with pathologically confirmed castration resistant metastatic prostate cancer who were treated with ^177^Lu-PSMA-617 during January 2017–February 2019. The characteristics of the patients are presented in Table 1. All men were docetaxel-resistant, except one patient who had received Estramustine, Cyclophosphamide, Etopocide and Ketoconazole. Prior to ^177^Lu-PSMA-617 treatment, most men had been treated with next-generation hormones such as abiraterone n = 6 (10%), enzalutamide n = 22 (35%), or both n = 28 (45%). Prior radium-223 dichloride treatment was in 16 (26%) and samarium-153 treatment in 6 (10%) patients. Other treatments included: Mitoxantrone, Carboplatin, Pembrolizumab, Vinorelbine, Gemcitabine, Oxaliplatin, Niraparib, Gefitinib, and Prosper study drug.

Responses to the ^177^Lu-PSMA-617 treatment were classified into four categories based on the PSA response to treatment: progression, PSA decrease <25%, <50% and >50%. Overall survival (OS) and PSA progression free survival (PFS) were determined. The median follow-up time was 1.4 years (IQR 0.5–2.2). The association of OS and PFS with pre-treatment staging PSMA-PET/CT imaging-derived metabolic tumor volume of metastases (MTV), tumor lesion activity (TLA), regional metastasis SUVmax values, pre-treatment PSA level, PSA-velocity, PSA doubling time and Gleason score were assessed. Possible side effects were reported either electronically using the Kaiku Health (Kaiku Health, Helsinki, Finland) communication system for self-reporting of adverse events or directly to the treating physician.

### 2.2. Tracer Production

The radiolabeled substances ^18^F-PSMA-1007, ^68^Ga-PSMA-11, ^177^Lu-PSMA-617 and ^68^Ga-PSMA-11 were provided by MAP Medical Technologies Oy under special licenses issued by the local regulatory authorities.

The ^18^F-PSMA-1007 was produced in one step on an automated synthesis module from commercially available PSMA-1007 precursor using a radiolabeling process in analogy to a published method [20]. The labelled product was diluted, sterilized by aseptic filtration through a 0.22 μm filter and dispensed by an automated dispenser under a grade A controlled environment. 

^68^Ga-PSMA-11 (^68^Ga-PSMA-HBED_CC) was prepared in analogy to a general method for preparing ^68^Ga-labeled DOTA-conjugated peptides directly from generator eluate [21]. ^68^Ga was obtained from a commercially available ^68^Ge/^68^Ga generator containing ^68^Ge on a silica gel modified with dodecyl gallate sorbent, and the radiolabeling was performed on a semi-automated synthesis module using commercially available PSMA-11 precursor.

Quality control of the drug products including sterility, radiochemical purity, radiochemical identity, endotoxin content and pH were tested by MAP Medical Technologies Oy, Tikkakoski Finland. The products were released by a qualified person after the results were shown to be compliant with the acceptance criteria of the recent European Pharmacopoeia standards.

### 2.3. PET-Imaging

A staging prostate specific membrane antigen (PSMA) PET/CT scan with either ^68^Ga-PSMA-1007 (n = 23) or ^18^F-PSMA-1007 (n = 39) was performed before ^177^Lu-PSMA-617 treatment. All PET/CT scans were carried out using a Siemens Biograph 6 TruePoint scanner (Siemens, Erlangen, Germany). Low-dose CT scans were acquired for attenuation correction and anatomic correlation. The CT acquisition parameters were: tube potential 130 kV and tube current, which was modulated using Care Dose 4D, typically between 5–100 mA (the quality reference was 40 mA). Both the extended FOV (700 mm) images for attenuation correction and the diagnostic FOV (500 mm) images for anatomic localization were reconstructed. 

PET scans were acquired in three-dimensional mode with 4 min/bed positions for the ^18^F/^68^Ga-PSMA scan. The imaging started 60 min after the tracer injection. The sinogram data were corrected for deadtime, decay and photon attenuation, and reconstructed in a 168 × 168 matrix. Image reconstruction was done using an ordered subset expectation maximization algorithm (3D-OSEM) with 4 iterations and 8 subsets, and the images were filtered with a 5 mm Gaussian kernel.

### 2.4. ^177^Lu-PSMA-617 Therapy

Eligibility for ^177^Lu-PSMA-617 therapy was based on staging PSMA PET/CT imaging. The dominant part of PSMA avid tumor volumes had activities at least 1.5 higher than those obtained physiologically by liver in PET/CT imaging. The ^177^Lu-PSMA-617 was labelled with non-carrier-added ^177^Lu and commercially available PSMA-617 precursor. The radiolabeling was performed in a one-step method using an automated synthesis module. The radiolabeled drug substance was isolated on a C18 cartridge and was formulated in an injections grade water solution containing ascorbic acid and ethanol immediately after elution from the cartridge. The solution was sterilized by aseptic filtration through a 0.22 um filter and dispensed in a grade A controlled environment. 

All patients received prophylactic treatment against nausea and their salivary glands were protected by cold pads following intravenous injection of ^177^Lu-PSMA-617. Treatment SPECT/CT scans were performed using a Siemens Symbia T2 2-headed SPECT/CT camera with 2-slice CT (Siemens, Erlangen, Germany) 24 h after every ^177^Lu-PSMA-617 injection.

### 2.5. Image Analysis

Image analysis was performed with Syngo.Via version VB30 MM Oncology application (Siemens Healthcare GmbH, Erlangen, Germany). Advanced workstation ADW, version 4.5 (General Electric, Boston, MA, USA) was used for tumor volume analysis when the number of metastases were >100.

The volume of interest (VOI) was placed in all PSMA-positive lesions with activities above the blood background and considered to be metastatic/cancerous in origin. Vendor-based automated lesion detection was used. A fixed SUVmax threshold of 2.8 was used. These lesions were manually corrected/adjusted for every patient. Total tumor burden was calculated from all patients by summing all of the PSMA avid tumor volumes from metastatic lymph nodes (separately pelvic, para-aortic and thoracic), bone metastases, possible liver and lung metastases and also activity within the prostate or prostate bed (if any) from staging ^18^F- or ^68^GA-PSMA PET/CT. PET/CT scanning was performed 24 d (IQR 19–34) prior to ^177^Lu-PSMA-617 treatment. Total metastatic tumor volume (MTVtotal) was assessed by combining all regional VOIs. MTV and total lesion activity (TLA) were defined separately from all regions as described above. TLA was calculated as MTV*average activity of the regional lesions (SUVmean). SUVmax was also defined from each metastatic region. 

### 2.6. Adverse Events and Remote Reporting via Kaiku

All patient-reported side effects were graded and recorded according to the Common Terminology Criteria for Adverse Events (CTCAE) Version 5.0, published: 27 November 2017. About one third (24/66) of patients chose to use the electronic Kaiku Health communication system for self-reporting of adverse events. Other patients reported side effects to their treating physician who recorded them using an electronic medical report after each treatment during clinical visits. 

### 2.7. Statistics

Results are reported as median with interquartile ranges (IQR). The Kaplan-Meier method was used to produce progression-free and overall survival curves for different PSA groups. Univariate Cox proportional hazard regression analyses were applied to determine the associations between MTV, TLA and SUVmax values (lymph nodes, prostate/prostatic fossa, and bone, liver and lung metastases) and both PFS and OS. All explanatory variables except for Gleason score, number of Lu-PSMA treatments and PSA response group underwent logarithmic transformation with base 2 prior to analysis such that the reported hazard ratios correspond to a doubling of the explanatory variable in question. Additionally, multivariable Cox regressions of progression-free and overall survival controlling for pre-treatment PSA were performed on para-aortic lymph node MTV and bone MTV.

## 3. Results

### 3.1. Pre-Treatment Disease Location and Tumor Burden

Metastatic disease in bones was found in 19% of the patients and metastatic disease in lymph nodes was found in 10% of the patients. Metastatic disease in both bones and lymph nodes was found in almost half of the patients (48%). Nine patients had less than six bone metastases, twelve had 6–20 bone metastases, fifteen had more than 20 bone metastases and nineteen had superscan-like bone metastases. Visceral metastases were found in 23% of patients (five in the lung and nine in the liver). Active disease in the prostatic fossa/prostate was found in fifteen (24%) patients. Total and regional tumor MTV, TLA and SUVmax values are presented in Table 2. The median percentage distribution of MTV was 80.5% (IQR 30–100) in bones and 9.2% (IQR 0–30) in lymph nodes. The largest volume of lymph node metastases, which had a median largest short-axis diameter of 1.3 cm (IQR 0.9–1.8), was observed in the para-aortic region.

### 3.2. ^177^Lu-PSMA-617 Therapies

All together, 171 ^177^Lu-PSMA-617 treatments were given. The median number of treatments per patient was three (IQR 3–5), and the range was 1–7. The treatment dose of 7081 MBq (IQR 6995–7188) was given in four-week intervals. The median interval between treatments was 33.5 d (IQR 27–58.5). Altogether, there were 33 patients that received 1–3 treatment cycles and 29 patients who received four or more treatment cycles. The highest number of treatment cycles was seven, which occurred in two subjects.

### 3.3. PSA and Treatment Response

The median PSA before treatment was 83.9 ng/L (IQR 13.0–305.8). The PSA velocity and doubling times were 6.5 ng/mL/months (IQR −1.6–32.9) and 2.5 months (IQR −0.5–4.2), respectively. A significant > 50% PSA decrease was observed in 58.7% of the patients (N = 37), a partial response (<50% but >25%) in 12.7% (N = 8), a minor response (<25%) in 12.7% (N = 8) and no response in 15.8% (N = 10). Any response was observed in a total of 53 patients (84%). The percentage change in the significant and partial treatment response groups was significantly higher than in other response groups (Figure 1). 

The median percentage decrease in PSA (all patients) was −19.8% (IQR −37.1–4.3) after the first treatment, −42.9% (IQR −69.8–16.5) after the second treatment and −70.4% (IQR −88.6–−35.1) after the third treatment. A trend towards a greater PSA decrease with a greater number of treatment cycles was also observed in patients who received more than three treatment cycles. For example, the median decrease in PSA was −94.2% (IQRT −91.0–−35.6) for patients who received six treatment cycles (n = 10). Altogether, there were six patients (10%) with an exceptional (more than 98%) PSA treatment response. The best response was a 99.9% PSA decrease, which represented a change from 121 to 0.1 ng/L. All of the patients with an excellent treatment response had detectable PSA levels after PSMA-RLT. In general, the PSA response was not associated with Gleason score, pre-treatment PSA kinetics, MTV, TLA, SUVmax values or lesion diameters. Figure 2 displays some examples of PSMA PET image pairs in four mCRPC patients with a significant reduction in PSA after PSMA-RLT.

### 3.4. Follow-Up and Survival

The median follow-up time was 1.7 years (y; IQR 0.7–2.2). The median PSA progression-free survival was 0.4 y (IQR 0.2–0.8) and the OS was 1.6 y (IQR 0.7–2.2). During follow-up, 36 (58%) patients died. The survival probability was significantly better among patients with significant (>50%) PSA response compared to non-responders, *p* < 0.04, Figure 3. The median OS in the best-response group was 1.9 y (IQR 1.2–2.3) compared to 0.5 y (IQR 0.3–1.5) in non-responders. 

OS was significantly associated with pre-treatment PSA, MTVtotal, TLAtotal, bone MTV, lymph node MTV and TLA, para-aortic (distant abdominal) lymph node MTV and TLA and number of Lu-treatments (Table 3). PFS was significantly associated with pre-treatment PSA, PSA velocity, MTVtotal and number of Lu-treatments (Table 4). In a multivariate analysis controlling for pre-treatment PSA, para-aortic MTV had an impact on OS (HR 1.51, CI 1.12–2.02, *p* = 0.006) and bone MTV had an impact on PFS (HR 1.27, CI 1.03–1.55, *p* = 0.02). 

### 3.5. Adverse Events

Before the first lutetium treatment, eight patients had grade 1 thrombopenia and one patient had grade 2 thrombopenia. In addition, four patients had grade 1 leucopenia and three patients had grade 2 leucopenia. Even though 85.8% of patients were anemic, forty-five patients had grade 1 hemoglobin values and eight patients had grade 2 hemoglobin values. Generally, changes in hemoglobin, white cell and platelet values were not treatment-limiting factors. 

In this patient cohort, grade 1–2 hemoglobin values were observed in 57 (91.9%) patients after treatment, but there were no grade 3 hemoglobin values. Grade 1–2 leucopenia was observed in thirteen patients and two patients had grade 3 leucopenia. Grade 1–2 thrombocytopenia was observed in seventeen patients and only one patient had grade 3 trombocytopenia.

Tiredness appeared to be the most serious patient-reported adverse event during treatment. Grade 3 tiredness was reported by ten (16.1%) and grade 1–2 tiredness by twenty-seven (43.5%) of the patients. Grade 3 pain was reported by four subjects (6.5%) and grade 3 infection was reported by two subjects. None of the grade 3 pain or infections were related to ^177^Lu-PSMA-617 treatment. Grade 1–2 dryness of the mouth was reported by thirty-three (53.2%) of the patients. Grade 1–2 nausea was reported by twenty-two (35.5%) and diarrhea by sixteen (25.8%) of the patients. Grade 4–5 side effects were not reported at all.

## 4. Discussion

^177^Lu-PSMA-617 treatment delivered in four-week intervals was safe and effective in terms of producing a significant PSA decrease in a substantial number of patients. Almost 60% of the patients had a significant PSA decline of more than 50%, which was associated with better OS and PFS. The median overall OS was 1.6 y. Although pre-treatment staging PSMA was not helpful in identifying responders and non-responders, MTVtotal provided prognostic information on OS and PFS. 

Cumulative knowledge and experience on PSMA-targeted RLT and its efficacy and safety has been accumulating over the last six years [1,2]. PSMA-RLT is becoming recognized as an option for advanced mCRPC patients, but optimal treatment protocols or the safety and efficacy of sequencing and combining PSMA-RLT with other treatments have not yet been established. Initial studies performed with a 2–8 GBq dose and six-, eight- or even up to twelve-week treatment intervals reported a >50% PSA response rate in 32–56% of patients and any PSA change in 50–91% of patients [5,6,7,8,9,14,16]. The reported effects on OS were 5–15.5 months. Later studies, which involved patient doses of 6 GBq and above and six-week treatment intervals [11,12,22,23], reported a >50% PSA response rate in 36–64% of patients, any PSA change in 71–97% of patients and OS of 10.7–13.3 months. Although these results overlap somewhat, it is tempting to think that shorter treatment intervals may broaden the therapeutic window and improve the outcome of RLT, especially among mCRPC patients with high volume disease and rapidly rising PSA. Moreover, it is known that some patients may require more than one treatment cycle to show a positive treatment response [19], which suggests that, especially in rapidly progressing mCRPC patients, a more aggressive approach may be needed.

Seifert et al. compared the safety and efficacy of two treatment regimens: 7.5 GBq every six weeks and 6 GBq every eight weeks [13]. Without a significant difference in safety, a higher dose with a shorter treatment interval yielded longer OS (12.7 vs. 11.3) and PFS (12.3 vs. 9.5), although these differences were not statistically significant. The rate of >50% PSA response rate also tended to be higher (54% vs. 35%). Furthermore, the VISION trial, which involved more than 800 patients and was performed with a fixed 7.4 GBq dose every six weeks, yielded a median OS of 15.3 months [17]. Besides our study, to our knowledge there is only one other retrospective study that considered a four-week treatment interval with fixed (7.4 GBq) treatment dose. The authors of the study reported a significant PSA decrease in 35% of patients and any decrease in 79% of patients, with an OS of over 2 y [24]. These results are similar to those from the present study, which observed a significant PSA response in 58% of the patients and any response in 84% of the patients. The lower OS (1.6 y) in our study is likely related to the presence of more advanced disease in our patient cohort. Our patient cohort contained a higher number of patients with liver metastases and a 21% higher median pre-treatment PSA-level. 

Although Rasul et al. reported a decrease in thrombocytes and leukocytes compared to baseline after the third treatment cycle, therapy was generally well tolerated by all of the patients, and no acute adverse effects or grade 4 hematological toxicity occurred [24]. Our results also showed this. In general, patients tolerated a four-week treatment interval with fixed dose well, and there were no grade 4–5 side effects. Almost 90% of our patients appeared to be anemic before PSMA-RLT due to disease involvement in bone marrow and prior treatments such as docetaxel, cabazitaxel, radium-223, samarium-53 and hormonal therapy. However, it is noteworthy that there was no significant worsening of bone marrow function with PSMA-RLT. On the contrary, it appeared that bone marrow deterioration was more likely related to the progression of the disease in the bone marrow. 

The observed grade 3 side effects (tiredness, pain, infection) were probably not related to the ^177^Lu-PSMA-617 treatment. Although tiredness was common, we think that fatigue is not a typical radioligand-related side effect, but is more probably associated with chemical castration, new generation hormone therapies, prior chemotherapy and extensive disease burden. Similarly, pain was not considered to be radioligand-associated but rather due to the presence of bone metastases. Among three patients, grade 3 severe pain was related to osteoporotic fractures. Mild nausea was quite common and was observed in 36% of cases despite prophylactic medication; however, in some cases, it seemed more likely to be related to obstipation. The side effects which were most likely related to ^177^Lu-PSMA-617 treatment include: dryness of the mouth, lips and eyes; nausea; diarrhea; fever during the first day after the infusion of the radioligand; impaired taste and smell; loss of appetite; infection; stomatitis; headache; dizziness; flulike symptoms; redness of the eyes; stiffness of muscles; insomnia; and tinnitus. 

The degree of PSA response was not significantly associated with pre-treatment PSA-kinetics, Gleason score, staging PSMA PET/CT imaging-derived tumor volumes, tumor volume activities or SUVmax values in metastatic regions. From a prognostic point of view, it was not surprising that high disease volume, reflected by a high pre-treatment PSA level, and liver metastases were both risk factors for lower OS in this study. The number of ^177^Lu-PSMA-617 treatments was also associated with better OS. This could be related to the fact that good partial responses encouraged oncologists to continue with treatment while non-responses contributed to treatment discontinuation. However, there were also high-volume metastatic patients who clearly benefited from serial treatments and experienced a continuous decline of PSA. Both any decline and a ≥50% PSA response in PSMA-RLT have been shown to be linked to better OS in a meta-analysis [1].

SUVmax values did not serve as a prognostic factor in our study. Other groups have similarly also reported that SUVmax values from PSMA-imaging were not significant prognostic factors for OS among patients treated with ^177^Lu-PSMA-617 [15,25]. Therefore, SUVmax is not a predictive biomarker for successful PSMA-RLT. However, to ensure good delivery of ^177^Lu-PSMA-617 radioligand to tumors, high PSMA uptake is to be expected in PET-imaging. Therefore, SUVmax values are used for the assessment of patient eligibility for PSMA-RLT, but the exact treatment threshold limits to be used are not known [3,11,15]. The low average PSMA expression seems to be linked to lower OS [26], but the lowest tolerable PSMA expression volume for PSMA-RLT is not known. 

Parameters for total tumor burden, MTVtotal and its activity, TLAtotal, bone MTV, lymph node MTV and TLA, especially para-aortic lymph node MTV and TLA, were significantly and negatively linked to OS. This highlights the importance of tumor volume control in patients with mCRPC. It is highly interesting that high volume metastatic lymph node involvement in the abdominal cavity and outside pelvic region had high independent risk for low OS. This finding was not explained by lymph node SUVmax, short axis diameter or possible additional visceral metastases. It is not known whether metastatic abdominal lymph nodes present more aggressive PCs per se or whether it is linked to surviving treatment-resistant cancer clones which possess a high risk for lower OS. Seifert al. have also shown that baseline PSMA tumor volume is negatively linked to OS [26]. They also investigated the treatment-related tumor volume reduction and found that it was correlated with OS only when patients with low PSMA expression were excluded [27]. Therefore, tumor volume alone might not be a suitable parameter to evaluate treatment responses in PSMA-RLT; instead, the balance between volumes of high and low PSMA uptake may be more suitable for this purpose [15]. In light of this, recent analyses from the phase 2 TheraP trial, showed that an SUVmean > 10 predicted a higher likelihood of a favorable PSMA-RLT response and that patients with an SUVmean less than 6.9 did not show any more superior response to cabacitaxel [28]. Other prognostic markers for poor OS among ^177^Lu-PSMA-617-treated mCRPC patients include: young age, low pre-therapeutic hemoglobin, a high number of platelets, high C-reactive protein, high lactate dehydrogenase, high levels of γ-glutamyl transferase, a high Gleason score, and regular need for pain medication [7,25,29]. FDG-positive tumor volume, mean intensity of PSMA avid tumor uptake, alkaline phosphatase, second line chemotherapy and visceral metastases have also shown to be prognostic biomarkers of OS [14,30].

This study has some limitations. Due to the retrospective analysis, this study is prone to selection bias. During the analysis period, the staging PSMA PET tracer was changed from ^68^Ga-PSMA-1007 to ^18^F-PSMA-1007. This could have potentially increased the variance of the measures of regional SUVmax. However, the impact on volumetric tumor burden analyses is likely to be small. As a result of technical, patient or treatment-related factors that prolonged individual treatment intervals, four subjects had a median treatment interval of more than 42 days. Nevertheless, the median treatment interval among all subjects was one month. In addition, ^18^F-FDG PET/CT scans were not made to assess eligibility for PSMA-RLT, which could have impacted the number of non-responders in this study. Moreover, it could have provided insight into why abdominal lymph node metastases were negatively associated with OS or whether any of the patients in this cohort had metastases with low PSMA uptake but elevated glucose metabolism. It is known that neuroendocrine differentiation can be present as an untreated primary pathology either together with prostatic adenocarcinoma or, more commonly, as a receptor inhibition resistance phenomenon in androgen deprivation therapy [31]. To investigate this possibility, ^18^F-FDG or ^68^Ga-Dotanoc tracers could be used for patients with mCRPC [32]. Recently, Buteau et al. showed in a phase 2 trial (TheraP) that a high FDG avid tumor volume (>200 mL) is a prognostic biomarker for lower treatment response regardless of whether patients were randomly assigned to treatment with ^177^Lu-PSMA-617 or cabazitaxel [28]. Therefore, ^18^F-FDG could be a potential tracer that can be used to identify mCRPC patients that could benefit from treatment intensification. In this regard, patient examples C and D in Figure 2, in which a significant PSA response was observed but OS remained below the median of the best response group, are especially interesting. Since traditional laboratory or imaging-related biomarkers are not able to distinguish ^177^Lu-PSMA-treatment responders from non-responders, better biomarkers are needed to aid in patient selection. One novel approach to identify potential biomarkers is to utilize liquid biopsy techniques in analyses of circulating tumor DNA and cells to measure the gene expression of PSMA, androgen receptors and their variants or neuroendocrine differentiation [33,34]. Since PSMA-RLT is relatively non-toxic, treatment combinations can be considered to further improve response rates. For example, PSMA-RLT with DNA damage response/replication stress response (DDR/RSR) repair inhibitors would be an interesting treatment strategy for mCRPC since activating these pathways can mediate resistance to PSMA-RLT [35].

## 5. Conclusions

^177^Lu-PSMA-RLT in a four-week interval with a fixed dose was safe and effective. A significant PSA response was observed in six out of ten patients, which was also associated with better OS and PFS. A shorter treatment interval may broaden the therapeutic window among mCRPC patients with high volume disease and rapidly rising PSA. Pre-treatment staging PSMA was not helpful in distinguishing responders from non-responders. However, parameters for tumor burden and its activity provide prognostic information related to OS and PFS. 

## Figures and Tables

**Figure 1 cancers-14-06155-f001:**
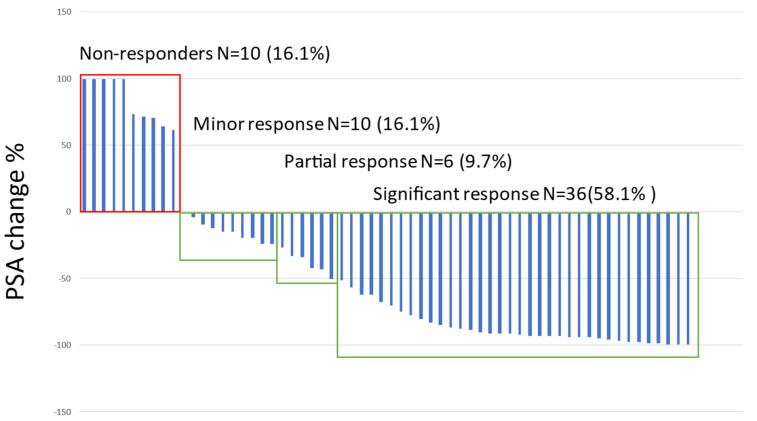
Waterfall plot on the best PSA response to ^177^Lu-PSMA-treatments.

**Figure 2 cancers-14-06155-f002:**
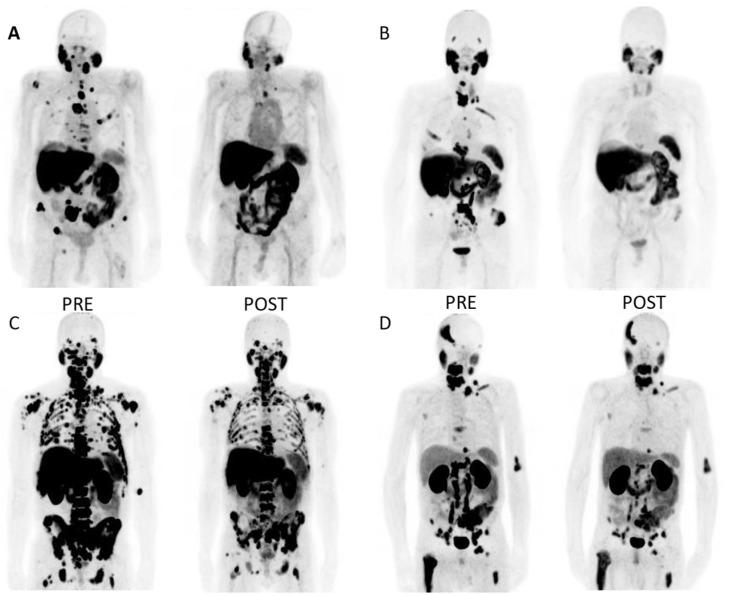
Example PSMA PET image pairs (**A**–**D**) in four mCRPC patients before and after PSMA-RLT. (**A**): Patient with bone metastases only. Pre-treatment PSA was 83.6 ng/L. He received seven cycles of ^177^Lu-PSMA-617, which resulted in a significant response. PSA after the treatments was 3.2 ng/L. PFS(PSA) was 23.5 months (mo) and OS was 2.4 y. (**B**): Patient with metastases mostly in bones but also in lymph nodes. Pre-therapy PSA was 263 ng/L. After four ^177^Lu-PSMA-617 cycles, a significant response was noted, and PSA declined to 6.5 ng/L. PFS was 7.8 mo and OS 2.3 y. (**C**): Patient with high volume bone metastases and low volume lymph node metastases. Pre-treatment PSA level was 926 ng/L. He received seven cycles of ^177^Lu-PSMA-617. PSA dropped to 54 ng/L. PFS was 4.8 mo and OS 1.0 y. (**D**): Large volume of metastatic lymph node involvement in abdomen and bone metastases. Pre-treatment PSA was only 9.1 ng/L, and the patient received four treatment cycles. PSA declined to 0.6 ng/L. The post-treatment images show a partial response in the abdominal lymph node metastases and clear treatment-refractory bone metastases with high PSMA activity. 4.5 mo later, biochemical relapse was noted. OS was 1.4 y.

**Figure 3 cancers-14-06155-f003:**
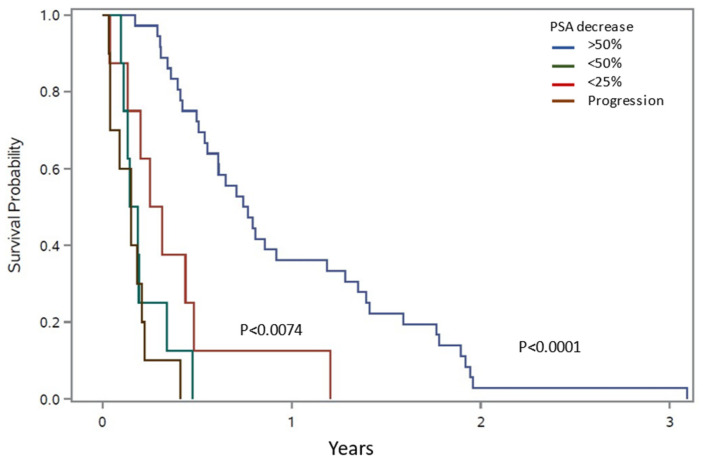
Survival potential (OS) according to different PSA response groups.

**Table 1 cancers-14-06155-t001:** Patient characteristics. Gleason score information was missing from two patients since a biopsy was not done due to a large degree of metastasis and clinically obvious T4N1M0 and T4N1M1 disease. T-stage information was missing from one subject.

Patients (n)	62
Age (y)	71.3 (IQR 66.7–75.4)
Years from diagnosis (y)	8.7 (IQR 4.3–13.2)
Original prostate cancer stage:	
Stage 1	4
Stage 2	3
Stage 3	21
Stage 4	33
Gleason score at diagnosis:	
6	8
7 (3 + 4)	9
7 (4 + 3)	9
8	15
9	18
10	1

**Table 2 cancers-14-06155-t002:** Metastatic tumor volumes (MTV), tumor lesion activity (TLA) and regional SUVmax values in all lesions, in bone, in lymph node (all), in pelvic lymph nodes, in abdominal lymph nodes, in thoracic lymph nodes, in prostate/prostatic fossa and in liver and lung metastases.

Tumor Burden and Tumor Lesion Activities	
Parameter	Median (IQR)
Metabolic tumor volume, all lesions (MTVtotal, cm^3^)	413.9 (IQR 68.9–1067.7)
MTV in bone metastases (cm^3^)	296.8 (IQR 47.7–965.1)
MTV in all lymph node metastases (cm^3^)	37.1 (IQR 14.8–112.9)
MTV in pelvic lymph node metastases (cm^3^)	22.9 (IQR 6.9–49.3)
MTV in para-aortic lymph node metastases (cm^3^)	35.3 (IQR 9.1–66.3)
MTV in thoracic lymph nodes	14.0 (4.1–41.1)
MTV in lung metastases (cm^3^)	1.7 (IQR 1.6–9.8)
MTV in liver metastases (cm^3^)	507.0 (IQR 204–622.6)
MTV in prostate/prostatic fossa (cm^3^)	17.9 (IQR 5.2–45.1)
Total tumor lesion activity (TLAtotal, cm^3^*SUVmax)	2786.5 (IQR 386.4–7977.6)
TLA in bone metastases	1725.4 (IQR 336.5–6699.2)
TLA in lymph node metastases	254.7 (IQR 336.5–6699.2)
TLA in pelvic lymph node metastases	191.9 (IQR 37.3–473.2)
TLA in para-aortic lymph node metastases	256.1 (IQR 38.3–736.9)
TLA in thoracic lymph nodes	52.3 (IQR 16.9–294.1)
TLA in lung metastases	6.7 (IQR 3.2–13.3)
TLA in liver metastases	4401.2 (IQR 2878.4–4589.7)
TLA in prostate/prostatic fossa	92.0 (IQR 7.9–17.6)
SUVmax in bone metastases	24.3 (IQR 9.7–42.3)
SUVmax in pelvic metastases	21.8 (IQR 9.2–32.0)
SUVmax in para-aortic metastases	19.7 (IQR 9–36.9)
SUVmax in thoracic metastases	9.5 (IQR 6.6.0–9.8)
SUVmax in lung metastases	3.0 (IQR 1.0–1.5)
SUVmax in liver metastases	16.8 (IQR 15.3–24.9)
SUVmax in prostate/prostatic fossa	12.8 (IQR 7.9–17.6)

**Table 3 cancers-14-06155-t003:** Possible predictors for overall survival (OS). MTV = Metastatic tumor volume, TLA = Total lesion activity (meanSUV*tumor volume).

Overall Survival (OS)		
Parameter	HR (95% CI)	*p*-Value
Years from diagnosis	0.85 (0.68–1.06)	0.144
PSA prior Lu-treatment	1.16 (1.03–1.30)	**0.018**
PSA velocity	1.13 (0.98–1.31)	0.102
PSA doubling time	1.09 (0.84–1.40)	0.523
Gleason score	1.03 (0.80–1.34)	0.801
Number of Lu-PSMA treatments	0.78 (0.63–0.97)	**0.028**
MTVtotal	1.24 (1.07–1.43)	**0.003**
TLAtotal	1.17 (1.03–1.32)	**0.017**
Bone MTV	1.15 (1.01–1.30)	**0.031**
Bone TLA	1.11 (0.99–1.23)	0.075
Bone SUVmax	0.94 (0.74–1.20)	0.609
Lymph node MTV total	1.26 (1.05–1.51)	**0.013**
Lymph node TLA total	1.17 (1.01–1.36)	**0.032**
Pelvic lymph node MTV	1.13 (0.92–1.37)	0.242
Pelvic lymph node TLA	1.09 (0.92–1.29)	0.325
Pelvic lymph node SUVmax	1.13 (0.80–1.60)	0.488
Para-aortic lymph node MTV	1.61 (1.21–2.12)	**0.0009**
Para-aortic lymph node TLA	1.37 (1.10–1.70)	**0.004**
Para-aortic lymph node SUVmax	1.26 (0.89–1.77)	0.195
Thoracic lymph node MTV	1.15 (0.95–1.40)	0.144
Thoracic lymph node TLA	1.08 (0.93–1.26)	0.298
Thoracic lymph node SUVmax	0.92 (0.63–1.35)	0.678
Liver MTV	1.43 (0.82–2.51)	0.210
Liver TLA	1.28 (0.78–2.10)	0.336
liver SUVmax	1.19 (0.32–4.44)	0.802
Lung MTV	0.70 (0.38–1.29)	0.255
Lung TLA	0.72 (0.37–1.43)	0.350
Lung SUVmax	0.35 (0.10–1.23)	0.101

**Table 4 cancers-14-06155-t004:** Possible predictors for PSA progression-free survival (PFS). MTV = Metastatic tumor volume, TLA = Total lesion activity (meanSUV*tumor volume).

PSA Progression-Free Survival (PFS)		
Parameter	HR (95% CI)	*p*-Value
Years from diagnosis	0.96 (0.81–1.14)	0.629
Pre-treatment PSA	1.12 (1.02–1.22)	**0.014**
PSA velocity	1.25 (1.09–1.43)	**0.001**
PSA doubling time	0.98 (0.78–1.22)	0.839
Gleason score	0.94 (0.78–1.14)	0.518
Number of Lu-PSMA treatments	0.79 (0.67–0.94)	**0.006**
MTVtotal	1.13 (1.00–1.26)	**0.037**
TLAtotal	1.08 (0.98–1.18)	0.119
Bone MTV	1.10 (0.99–1.21)	0.069
Bone TLA	1.07 (0.98–1.16)	0.157
Bone SUVmax	0.92 (0.75–1.13)	0.438
Lymph node MTV	1.05 (0.92–1.20)	0.443
Lymph node TLA	1.04 (0.94–1.15)	0.481
Pelvic lymph node MTV	1.05 (0.90–1.23)	0.558
Pelvic lymph node TLA	1.04 (0.91–1.18)	0.556
Pelvic lymph node SUVmax	1.20 (0.89–1.61)	0.229
Para-aortic lymph node MTV	1.07 (0.86–1.32)	0.541
Para-aortic lymph node TLA	1.04 (0.89–1.21)	0.656
Para-aortic lymph node SUVmax	1.07 (0.79–1.46)	0.648
Thoracic lymph node MTV	1.02 (0.88–1.19)	0.763
Thoracic lymph node TLA	1.00 (0.89–1.14)	0.936
Thoracic lymph node SUVmax	0.88 (0.63–1.21)	0.425
Liver MTV	1.38 (0.82–2.32)	0.223
Liver TLA	1.27 (0.79–2.06)	0.328
Liver SUVmax	1.11 (0.32–3.88)	0.870
Lung MTV	0.78 (0.51–1.21)	0.271
Lung TLA	0.82 (0.50–1.34)	0.419
Lung SUVmax	0.85 (0.44–1.64)	0.621

## Data Availability

The data presented in this study are available on request from the corresponding author.

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
