# Peer review of "Single Center Experience with a 4-Week 177Lu-PSMA-617 Treatment Interval in Patients with Metastatic Castration-Resistant Prostate Cancer"

_cancers, 2022, doi:10.3390/cancers14246155_

Round 1

Reviewer 1 Report

This study reports a retrospective analyses of metastatic CRPC patients at a single center that underwent 177Lu-PSMA-617 therapy at two different doses for 4 week interval. Significant PSA responses were observed in 58.7% of patients that correlated with overall survival and progression free survival (PFS). They report that PFS correlate significantly with pre-tretament PSA, PSA velocity, metastatic total volume and number of Lu-treatments. There are limitations in the study. It is a retrospective analyses where the eligibility for PSMA-RLT was not assessed. THis could impact non-responders reported in the study. In view of these limitations, this study does not provide meaningful new data to draw meaningful conclusions. 

Minor: Gleason 10- how many patients were included? This information is missing. 

Line 209 Page 6: Section on therpies; Please correct the number of treatments. 

Author Response

Reviewer 1:

This study reports a retrospective analyses of metastatic CRPC patients at a single center that underwent 177Lu-PSMA-617 therapy at two different doses for 4 week interval. Significant PSA responses were observed in 58.7% of patients that correlated with overall survival and progression free survival (PFS). They report that PFS correlate significantly with pre-tretament PSA, PSA velocity, metastatic total volume and number of Lu-treatments. There are limitations in the study. It is a retrospective analyses where the eligibility for PSMA-RLT was not assessed. THis could impact non-responders reported in the study. In view of these limitations, this study does not provide meaningful new data to draw meaningful conclusions. 

  • Thank you for comments. As a retrospective study there is surely some limitations we have to address and discuss. We did not do FDG scan for these patients. We added discussion about FDG PET/CT and neuroendocrine differentiation among ADT treated patients: lines 462-477:
    • Moreover, it could have provided insight into why abdominal lymph node metastases were negatively associated to OS or whether any of patients in this cohort had metastases with low PSMA uptake but elevated glucose metabolism. It is known that neuroendocrine differentiation can be present as untreated primary pathology together with prostatic adenocarcinoma or more commonly as receptor inhibition resistance phenomenon in androgen deprivation therapy[33]. To investigate such possibility 18F-FDG or 68Ga-Dotanoc tracers could be used for patients with mCRPC[34]. Recently, Buteau et al. showed in phase 2 trial (TheraP) that high volume of FDG avid tumor volume, > 200ml, is prognostic biomarker for lower treatment response regardless of randomly assigned treatment, ¹⁷⁷Lu-PSMA-617 vs. cabazitaxel[30]. Therefore, 18F-FDG could be potential tracer to identify mCRPC patients that could especially benefit from treatment intensification. In this regard patient examples C and D in figure 2 (Please, see page 10, suggested by reviewer 3) are especially interesting since significant PSA response was observed but OS remained below the median of the best response group.
  • Also, you raised concern about PSMA-RLT eligibility. This was assessed with PSMA PET/CT prior treatment. Following sentence is now added to methods section, in chapter 177Lu-PSMA-617 therapy:
    • Eligibility for 177Lu-PSMA-617 therapy was based on staging PSMA PET/CT imaging. Dominant part of PSMA avid tumor volumes had activities at least 1.5 higher than that of liver PET/CT imaging.
    • Also, by checking Table 2 (former Table 1) you could see the high median SUVmax values for bone, liver and abdominal lymph node metastases.

Minor: Gleason 10- how many patients were included? This information is missing. 

  • This typo is now corrected. We added a table where patients characteristics is displayed

Line 209 Page 6: Section on therpies; Please correct the number of treatments. 

  • The number of therapies is 171. There were cases with 1-2 therapies, median being 3 therapies/patient

Reviewer 2 Report

Line 72 - Define RLT in line 72 rather than line 80

Line 95 – Clarify number of GS patients

Line 102-104 – sentence needs finishing and is not complete

Line 196 – its very late to be introducing a new abbreviation of patients given that words has been used in full through the whole article

Line 198 – sentence does not make sense re: superscan

It will be important to mention (and provide discussion of) this very recent new publication relating to biomarkers in pts treated with Lu PSMA in Lancet Oncology - https://www.sciencedirect.com/science/article/abs/pii/S1470204522006052?dgcid=author

Author Response

Reviewer 2:

Thank you for your specific comments. I have replied the subsequent change below each of your suggestion.

Line 72 - Define RLT in line 72 rather than line 80

  • Good point. PSMA-RLT is defined now already in line 69:
    • It is known that some mCRPC patients who do not respond to the first PSMA targeted radioligand therapy (PSMA-RLT) might respond after subsequent 1-2 treatments [19].

Line 95 – Clarify number of GS patients

  • We added a Table 1 where number of missing GS10 patients is mentioned too

Line 102-104 – sentence needs finishing and is not complete

  • Sentence lacked verb and has been modified as follows:
    • Association of OS and PFS to pre-treatment staging PSMA-PET/CT imaging derived metabolic tumor volume of metastases (MTV), tumor lesion activity (TLA), regional metastasis SUVmax values, pre-treatment PSA level, PSA-velocity, PSA doubling time and Gleason score were assessed.

Line 196 – its very late to be introducing a new abbreviation of patients given that words has been used in full through the whole article

  • I agree. Now this abbreviation of patients has been removed and the start of this chapter has been modified as follows:
    • Metastatic disease in bones only was found in 19% and in lymph nodes only in 10% of the patients. Combination of these was found in almost half of the patients (48%). 9 patients had less than six bone metastases, 12 had 6-20 bone metastases, 15 had more than 20 bone metastases, but less than superscan appearance and superscan like bone metastases were observed in 19 patients. Visceral metastases were found in 23% of patients (5 lung and 9 liver).

Line 198 – sentence does not make sense re: superscan

  • Sentences containing superscan have been modified to make these more reader friendly. Please, see above.

It will be important to mention (and provide discussion of) this very recent new publication relating to biomarkers in pts treated with Lu PSMA in Lancet Oncology - https://www.sciencedirect.com/science/article/abs/pii/S1470204522006052?dgcid=author

  • Thank you for bringing this up, important thing. This has been now added to discussion lines 442-445:
    • In light of this, the recent analyses of TheraP, phase2 trial, showed that SU-Vmean > 10 was predictive for higher likelihood for favorable PSMA-RLT response and patients with SUVmean less than 6.9 did not show any more su-perior response to cabacitaxel [30].

And also lines 462-477:

          Moreover, it could have provided insight into why abdominal lymph node metastases were negatively associated to OS or whether any of patients in this cohort had metastases with low PSMA uptake but elevated glucose metabo-lism. It is known that neuroendocrine differentiation can be present as un-treated primary pathology together with prostatic adenocarcinoma or more commonly as receptor inhibition resistance phenomenon in androgen depriva-tion therapy[33]. To investigate such possibility 18F-FDG or 68Ga-Dotanoc trac-ers could be used for patients with mCRPC[34]. Recently, Buteau et al. showed in phase 2 trial (TheraP) that high volume of FDG avid tumor vol-ume, > 200ml, is prognostic biomarker for lower treatment response regardless of randomly assigned treatment, ¹⁷⁷Lu-PSMA-617 vs. cabazitaxel[30]. There-fore, 18F-FDG could be potential tracer to identify mCRPC patients that could especially benefit from treatment intensification. In this regard patient exam-ples C and D in figure 2 (Please see page 10, suggested by reviewer 3) are especially interesting since significant PSA response was observed but OS remained below the median of the best response group.

also sentence in 480-483. was modified:

          One novel approach to identify potential biomarkers is to utilize liquid biopsy techniques in analyses of ctDNA and circulating tumor cells for gene expres-sion of PSMA, androgen receptors and its variants or neuroendocrine differen-tation [35,36][35]

Reviewer 3 Report

Brief Summary: The aim of the study by Kemppainen et al., was to assess the impact of a 4-week interval 177Lu-PSMA-671 treatment on metastatic castration resistant prostate cancer patients. Moreover, the authors sought to investigate whether the assessment of clinical parameters, such as pre-treatment PSMA PET/CET tumor staging, may prognosticate treatment response to a 4-week interval 177Lu-PSMA-671 intervention. They show that a 4-week interval 177Lu-PSMA-671 treatment of patients with metastatic castration resistant tumors (both primary and metastases) significantly improves PSA responses and correlates with better overall and progression-free survival in more than half of the assessed cohort. However, no significant correlation was observed between pre-treatment parameters and response to 177Lu-PSMA-671 intervention, warranting further investigation for better patient stratification. Overall, this is a well-written study that presents clinically-relevant and interesting data on a more intensive application of 177Lu-PSMA-671 treatment for the metastatic castration resistant prostate. However, the manuscript would benefit from some additional analyses/clarifications to be considered for publication.

Strengths of the study: 

·       Assessment of alternative, more intensive 177Lu-PSMA-671 treatment regimen in terms of therapeutic efficacy and safety.

·       Independent patient cohorts of metastatic castration resistant prostate cancer patients.

Weaknesses of the study:

·       Use of different PSMA PET/CT imaging agents for pre-treatment staging may hinder reliability of prognostic value (which the authors adequately discuss in their study).

·       Lack of more in-depth analyses and discussion on the differential treatment response between patients (non- responders vs excellent responders)

Comments:

Introduction: This section presents background information on 177Lu-PSMA-671 treatment and metastatic castration resistant prostate cancer, with clear aims of the study. Comments:

Materials and Methods: The experimental procedures and study design are comprehensively explained. Comments:

·       In the “subjects and study design” section, the authors should provide a table with the patient characteristics for a clearer understanding of the cohort used. [minor]

·       In section “Image analyses”, the authors outline the process for acquisition and analysis of PET/CT scans but they show no representative images. It would be extremely useful to add a figure or figure panels of representative PSMA PET/CT scans of responders and non-responders (linked to Figure 1). [minor]

Results and Figures: Overall, the results and figures are described and presented well but could use some improvements as also pointed above. Comments:

·       How were “excellent” responders classified? Where there any “exceptional” responders? From Figure 1, it is evident that some patient had an 100% drop in their PSA. The authors should clarify this and further discuss it. [major]

·       The authors could point out and discuss whether patient with histologically different metastatic castration resistant prostate cancer, e.g. NEPC, if any, had different 177Lu-PSMA-671 treatment responses and comment on whether patient stratification based on histological subtype would make sense in this setting. [major]

Discussion and Conclusions: The authors present their findings, discuss them with regards to published literature and point out the study’s limitations. This section would benefit from some further discussion on the points mentioned above.

Author Response

Thank you for your thorough review and comments. Based on these comments we think we were able to significantly improve this manuscript. We have done the following based on your suggestions:

  • Added table 1. for patient characteristics
  • Added representative images, Figure 2, on page 10.
  • Excellent responders were originally classified as psa change >50%. As this kind of change is considered more like significant response based on large prospective trials. Therefore, I changed term excellent to significant (which I think would be more appropriate) and psa change <50 but >25% is considered now partial response. Figure 1 is changed accordingly.
  • Exceptional responders, PSA change close to 100% (>98%) is now mentioned in results section, lines 253-256: There were all together 6 patients (10%) with exceptional, more than 98% treatment response in PSA. Best response was 99.9% change from 121 to 0.1. (Unfortunately, we don´t have post scan from this patient since his follow-up was continued in other country). All patients with excellent response had detectable PSA levels after PSMA-RLT.
  • NEPC and FDG is now addressed in discussion, lines 462-477:

Moreover, it could have provided insight into why abdominal lymph node metastases were negatively associated to OS or whether any of patients in this cohort had metastases with low PSMA uptake but elevated glucose metabo-lism. It is known that neuroendocrine differentiation can be present as un-treated primary pathology together with prostatic adenocarcinoma or more commonly as receptor inhibition resistance phenomenon in androgen depriva-tion therapy[33]. To investigate such possibility 18F-FDG or 68Ga-Dotanoc trac-ers could be used for patients with mCRPC[34]. Recently, Buteau et al. showed in phase 2 trial (TheraP) that high volume of FDG avid tumor vol-ume, > 200ml, is prognostic biomarker for lower treatment response regardless of randomly assigned treatment, ¹⁷⁷Lu-PSMA-617 vs. cabazitaxel[30]. There-fore, 18F-FDG could be potential tracer to identify mCRPC patients that could especially benefit from treatment intensification. In this regard patient examples C and D in figure 2 are especially interesting since significant PSA response was observed but OS remained below the median of the best response group.

and also sentence 482-483 was modified: One novel approach to identify potential biomarkers is to utilize liquid biopsy techniques in analyses of ctDNA and circulating tumor cells for gene expression of PSMA, androgen receptors and its variants or neuroendocrine differentation [35,36]

Round 2

Reviewer 1 Report

Authors have addressed raised concerns in the revised manuscript 

Reviewer 3 Report

The authors have addressed all the comments and the manuscript has improved.